# Electrochemical C–N bond activation for deaminative reductive coupling of Katritzky salts

Yeqing Liu[1,3], Xiangzhang Tao[1,3], Yu Mao[1], Xin Yuan[2], Jiangkai Qiu [2], Linyu Kong[1], Shengyang Ni[1✉], Kai Guo [2✉], Yi Wang [1✉] & Yi Pan[1]

Electrosynthesis has received great attention among researchers in both academia and industry as an ideal technique to promote single electron reduction without the use of expensive catalysts. In this work, we report the electrochemical reduction of Katritzky salts to alkyl radicals by sacrificing the easily accessible metal anode. This catalyst and electrolyte free platform has broad applicability to single electron transfer chemistry, including fluor-oalkenylation, alkynylation and thiolation. The deaminative functionalization is facilitated by the rapid molecular diffusion across microfluidic channels, demonstrating the practicality that outpaces the conventional electrochemistry setups.

[1] Jiangsu Key Laboratory of Advanced Organic Materials, State Key Laboratory of Coordination Chemistry, School of Chemistry and Chemical Engineering, Nanjing University, 163 Xianlin Avenue, Nanjing 210023, China. [2] College of Biotechnology and Pharmaceutical Engineering, Nanjing Tech University, 30 Puzhu Road South, Nanjing 211816, China. [3] These authors contributed equally: Yeqing Liu, Xiangzhang Tao. ✉email: nishengyanghi@qq.com; kaiguo@njtech.edu.cn; yiwang@nju.edu.cn

Electrochemical synthesis has experienced a renaissance in academic research and industry due to its sustainable and practical nature[1,2]. Numerous electrochemical approaches have been disclosed to access economic, scalable, and unique transformations triggered by electricity[3–5]. Rather than chemical oxidants or reductants, electrochemistry utilizes electrons and electron holes to achieve atom-economy and reduce carbon emissions. Additionally, the cheap and accessible metal anodes can be sacrificed to accomplish single electron reduction for the formation of radical intermediates. The broad substrate scope and mild cell conditions enable the wide application of such processes. For instance, Loren reported the reductive cross-coupling reaction of NHPI esters and organic halides with sacrificial Zn anode (Fig. 1A)[6]. Mei and Rueping independently described a nickel-catalyzed electrochemical reductive relay cross-coupling of organic bromides with sacrificial iron anode[7,8]. Ackermann also realized the C-H alkylation of unactivated 8-aminoquinoline amides by sacrificial zinc anode[9].

The concise electrosynthesis of C-C bonds from sp[3]-hybridized carbon radical represents significant synthetic values and inevitable challenges[10–12]. Followed by electron-promoted decarboxylative cross-coupling of redox-active esters (Fig. 1a)[6], we recently realized tertiary alcohol-derived alkyl carbazates for deoxygenative functionalization of heteroarenes under mild electrolytic conditions (Fig. 1b)[13]. The effort in search of other readily accessible radical precursors is still highly appreciated. Primary amine-derived Katritzky pyridium salts have been employed as carbon radical surrogates for transition-metal catalysis[14,15] and photoredox chemistry[16,17]. Previous approaches on reductive cross-coupling of Katritzky salts required expensive photocatalysts[18], electron donor-acceptor (EDA) complexes[19] or heavy metal catalysts[20–23] for the single electron transfer process. The analogous SET outcome could be achieved by cathodic reduction without chemical catalyst loading[24]. Meanwhile, Katritzky salts could also serve as the electrolytes under the cell conditions. Furthermore, recent advances in microfluidic techniques have offered rapid and controllable species transport within a micrometer channel for improved electrolytic performance[25].

In this work, we report an electrochemical reductive deaminative cross-coupling of Katritzky salts with various radical acceptors (Fig. 1c). The continuous-flow technology with large contact area of electrodes effectively accelerated the reaction rate.

## Results

**Reaction optimization.** We began our study with evaluation of the conditions for this envisioned electrochemical reductive cross-coupling reaction between pyridinium tetrafluoroborate **1** derived from cyclohexylamine and α-trifluoromethyl alkene **2**. After careful investigation of the reaction conditions, the cross-coupling product **3** could afford in DMSO with a 84% GC yield (79% isolated) by using zinc as the sacrificial anode and nickel foam cathode in a undivided cell under 5.0 mA constant current for 4 h at room temperature (Fig. 2, entry 1, with 1.7 equiv. of triphenylpyridine isolated). Additional electrolytes such as n-Bu₄NPF₆ are not necessary for this reaction (entry 2). Other solvents such as DMA, DMF, NMP, and MeCN resulted in diminished yields (entry 3–6). When switching the sacrificial anode to carbon anode, the reaction was suspended (entry 7). Other sacrificial metal such as Fe afforded a moderate yield (entry 8). Changing the nickel foam cathode to carbon cathode, an acceptable yield can still be obtained (entry 9). Increasing the current and reducing the time could not improve the yield (entry 10). When conducted under air, the reaction yield dropped significantly (entry 11). Without electricity, the reaction was terminated (entry 12). Using zinc powder instead of electricity, the reaction effiency was decreased and 20% yield was obtained (entry 13).

**Substrate scope.** With the optimized conditions in hand, the generality of this electrochemical reductive coupling reaction has been evaluated. A range of secondary Katritzky salts were verified for this electrolytic protocol (Fig. 3). Katritzky salts of cyclic primary amines (**3–5**) proceeded smoothly under the optimal reaction conditions providing the cross-coupled products in moderate to excellent yields. Katritzky salts containing cycloether (**6**, **7**), NBoc (**8**) could afford the fluoroalkenylated products in good yields. Linear and branched alkylamines (**9–12**) also could be converted to the products. Large ring (**13**) could also be tolerated. Katritzky salts containing hydroxyl groups (**14–16** and **21**) could apply to this reaction. When using the Katritzky salt of amino acid, the corresponding deaminative coupling product (**20**) could be obtained. Derivatives of alogliptin (**23**) and iso-xepac (**24**) were compatible to afford the corresponding

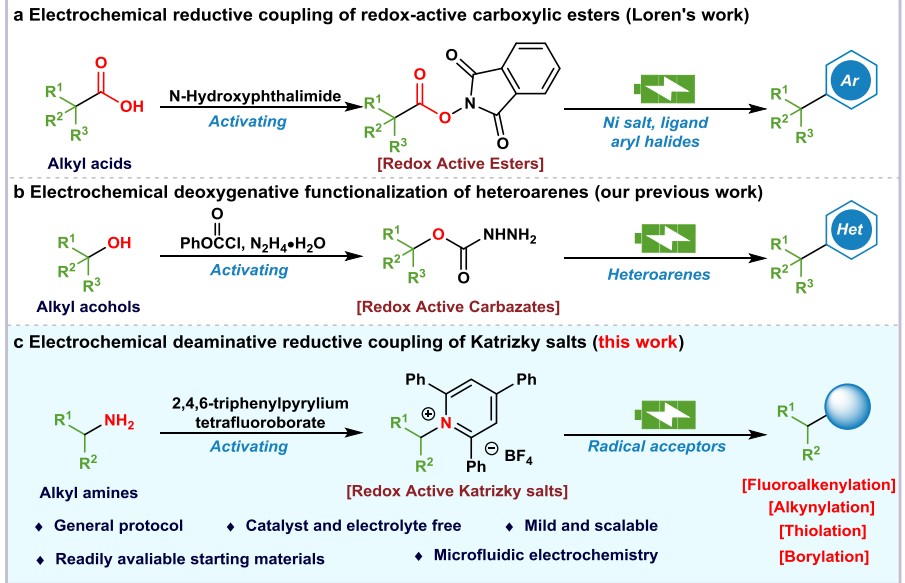

**Fig. 1 Electrochemical cross-coupling reactions of activated carbon radicals. a** Decarboxylative arylation. **b** Deoxygenative arylation. **c** Deaminative functionalization.

| Entry | Deviation from standard conditions | Yield % |
|---|---|---|
| 1 | none | 84[b] (79)[c] |
| 2 | nBu$_4$NBF$_4$ (0.1 M) | 77 |
| 3 | DMA instead of DMSO | 68 |
| 4 | DMF instead of DMSO | 62 |
| 5 | NMP instead of DMSO | 73 |
| 6 | MeCN instead of DMSO | 21 |
| 7 | C anode instead of Zinc anode | trace |
| 8 | Fe anode instead of Zinc anode | 68 |
| 9 | C cathode instead of NFE cathode | 70 |
| 10 | 10 mA, 2h | 76 |
| 11 | under air | 35 |
| 12 | no electicity | trace |
| 13 | Zn powder[d], no electricity | <20 |

[a] Reaction conditions: 1 (0.2 mmol), 2 (0.1 mmol), in 4.0 mL DMSO at room temperature for 4 h. I = 5.0 mA. Zinc anode, NFE cathode, under argon. [b] GC yield using decane as internal standard. NFE = nickel foamed electrode. [c] isolated yields. [d] Znic powder in 300-400 mesh was used.

**Fig. 2 Optimization of the reaction conditions.** Deviation from the standard conditions including electrolyte, solvent, electrode, current, additive and the corresponding yields of the coupling product.

difluoroalkenes in moderate yields (Fig. 3a). Subsequently, different substituted α-trifluoromethyl alkenes were assigned to the standard conditions. Trifluoromethylated alkenes (25, 26) containing methoxy-phenyl groups furnished the target products in good yields. Styrenes with electron-withdrawing substitution such as ester (27), CF$_3$ (28), CN (29) and F (30) could proceed with excellent yields (71%-82%). Olefin (31) or hydroxyl (32) containing substrates furnished the target products in good yields (Fig. 3b). Furthermore, this electrochemical reductive cross-coupling platform was also applied in the late-stage modification of biologically active molecules. The corresponding difluoroalkene derivatives of fenbufen (33), indomethacin (34), and estrone (35) were afforded in high efficiencies (Fig. 3c).

After establishing the reaction setups, we explored other radical receptors under this framework (Fig. 4). Sulfur-containing motifs are frequency present in nature products, bioactive molecules, and asymmetric catalysis as chiral catalysts/ligands[26,27]. The C(sp$^3$)-S bond construction has been a long-term focus in our laboratory[28,29]. We chose p-tolyl disulfide as the radical acceptor, after switching the ratio of the two starting materials, it was found that under the standard condition, the corresponding target product 36 can be obtained in 57% yield. Substrates with either electron-withdrawing or electron-donating tethers could afford the corresponding sulfide compounds (37, 38). Br or Cl substituted disulfides resulted in 39 and 40 in moderate yields. Other secondary Katritzky salts can also afford the target products (41–44). However, the primary Katritzky salts were unable to participate in the reaction. Notably, under the same conditions, diselenide can be converted into corresponding alkyl substituted selenide 45 in 58% yield (Fig. 4a).

The alkynylation of sp$^3$ carbon centers has significant interest and attracted extensive attention of synthetic chemists[30,31] Alkynyl p-tolylsulfones are served as suitable radical acceptor in recent reports[32,33]. We envisage to merge the electrochemical reductive strategy with alkynylating procedure to forge the C(sp$^3$)-C(sp) bond (Fig. 4b). By slightly modification of the standard conditions (DMF as solvent and Fe anode), alkynyl p-tolylsulfone could react smoothly with pyridinium salt of 4-piperidinamine to afford the corresponding product 46 in 54% yield. Methyl substitution in the para- or ortho-position could not affect the reaction (47, 48). Notably, bromine substitution on alkynyl p-tolylsulfone can be

retained in the reaction with 57% isolated yield (49). The cyclohexanamine was also tolerated to obtain the target product (50). To further demonstrate the utility of the electrochemical cross-electrophile coupling reaction, we examined the applicability of other radical receptors (Fig. 4c). We first used difluorovinylbenzonitrile (21) as radical receptor, the coupling product (52) was furnished in 25% yield. Using ethyl 2-((phenylsulfonyl)methyl) acrylate (53) as radical acceptor, the target product (54) was obtained in 37% yield. With B$_2$Cat$_2$, the borylated products (55–59) were isolated in good yields. In addition, other alkyl sources with trifluoroethylene were examined under similar conditions (Fig. 4d). First we used cyclohexyl iodide (60), which can be successfully converted to the target product 3. Redox-active NHPI ester (61) can also be successfully transformed into the same product under similar electrochemical conditions. Considering the potential improvement of the reaction efficiency, we extended the reaction to microfluidic electrochemistry platform (Fig. 4e). The merge of continuous flow technique and electrochemistry offers great opportunity for finely tuning the reaction parameters, expanding the contact area of electrodes and enhancing the reproducibility over the batch reactor. After brief optimizations, the target product 49 could be obtained in 25% yield at 10.0 mA in 600 seconds using SS304 steel anode and platinum cathode. The rapid molecular diffusion across microfluidic channels demonstrated the practicality that outpaces the conventional electrochemistry setups. To demonstrate the synthetic potential of this reductive cross-coupling, a gram scale reaction was performed with 1 and α-trifluoromethyl alkene 2 under the modified conditions (I = 50.0 mA, 8 h). The difluoroalkene product 3 was obtained in 57% yield (Fig. 4f). The radical trapping experiment with TEMPO resulted in no desired alkene product 3, and only the cyclohexyl adduct 64 was detected by GC-MS, which elucidates a radical pathway involved in this process (Fig. 4g)[34].

## Discussion

In conclusion, a catalyst and electrolyte free electrochemical reductive deaminative cross-coupling is described. This anodic sacrificing approach enables the traceless electrical C-N activation of Katritzky salts derived from ubiquitous primary amines, featuring the cost-effective apparatus, broad substrate scope, and high chemoselectivity for the incorporation of sp$^3$ carbon radicals into various organic moieties under mild cell conditions.

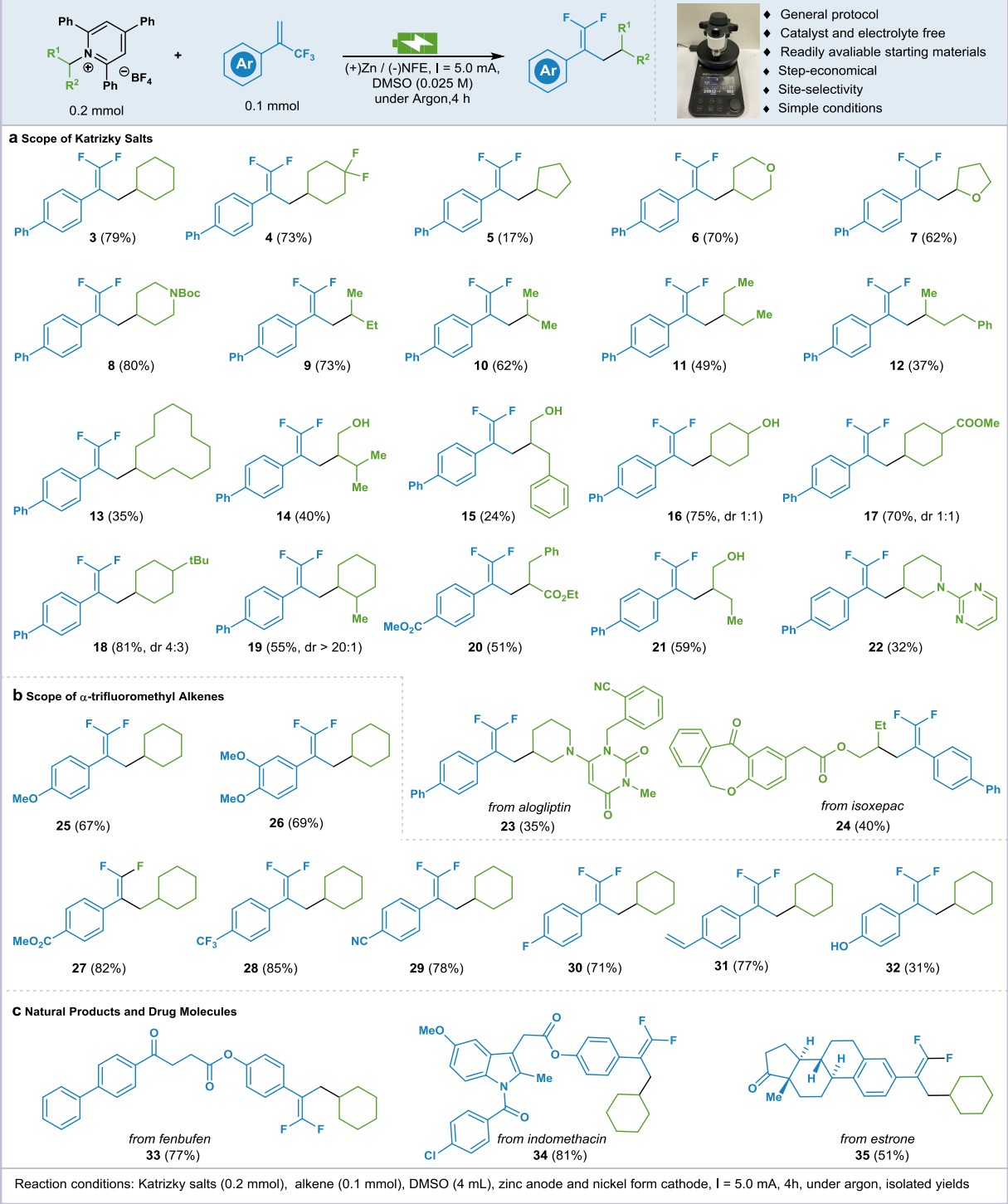

**Fig. 3 Scope of the electro-reductive cross-coupling reaction of Katritzky salts with α-trifluoromethyl alkenes. a** Scope of Katritzky salts. **b** Scope of alkenes. **c** Natural products and drugs.

## Methods

**General information**. All commercial reagents were used without additional purification unless otherwise specified. Solvents were purified and dried according to standard methods prior to use. All reactions were carried out under a nitrogen atmosphere with dry, freshly distilled solvents under anhydrous conditions, unless otherwise noted. Silica gel column chromatography was carried out using silica Gel 60 (230–400 mesh). Analytical thin layer chromatography (TLC) was done using silica Gel (silica gel 60 F254). TLC plates were analyzed by an exposure to ultraviolet light. NMR experiments were measured on a Bruker AVANCE III-400 or 500 spectrometer and carried out in deuterated chloroform (CDCl₃) ¹H NMR and ¹³C NMR spectra were recorded at 400 MHz or 500 MHz and 100 MHz or 125 MHz spectrometers, respectively. ¹⁹F NMR spectra were recorded at 376 MHz or 470 MHz spectrometers. Chemical shifts are reported as δ values relative to internal TMS (δ 0.00 for ¹H NMR), chloroform (δ 7.26 for ¹H NMR), chloroform (δ 77.16 for ¹³C NMR). The following abbreviations are used for the multiplicities: s: singlet, d: doublet, dd:doublet of doublet, t: triplet, q: quadruplet, m: multiplet, br: broad signal for proton spectra; Coupling constants (J) are reported in Hertz (Hz). Melting points were recorded on a Fisher-Johns 12-144 melting point apparatus and were uncorrected. HRMS were recorded on a Bruker microTOF-Q111. GC-MS spectra were performed on Shimadzu QP2010 (EI Source). In a general experiment, a borosilicate glass tube was used as a reaction tube.

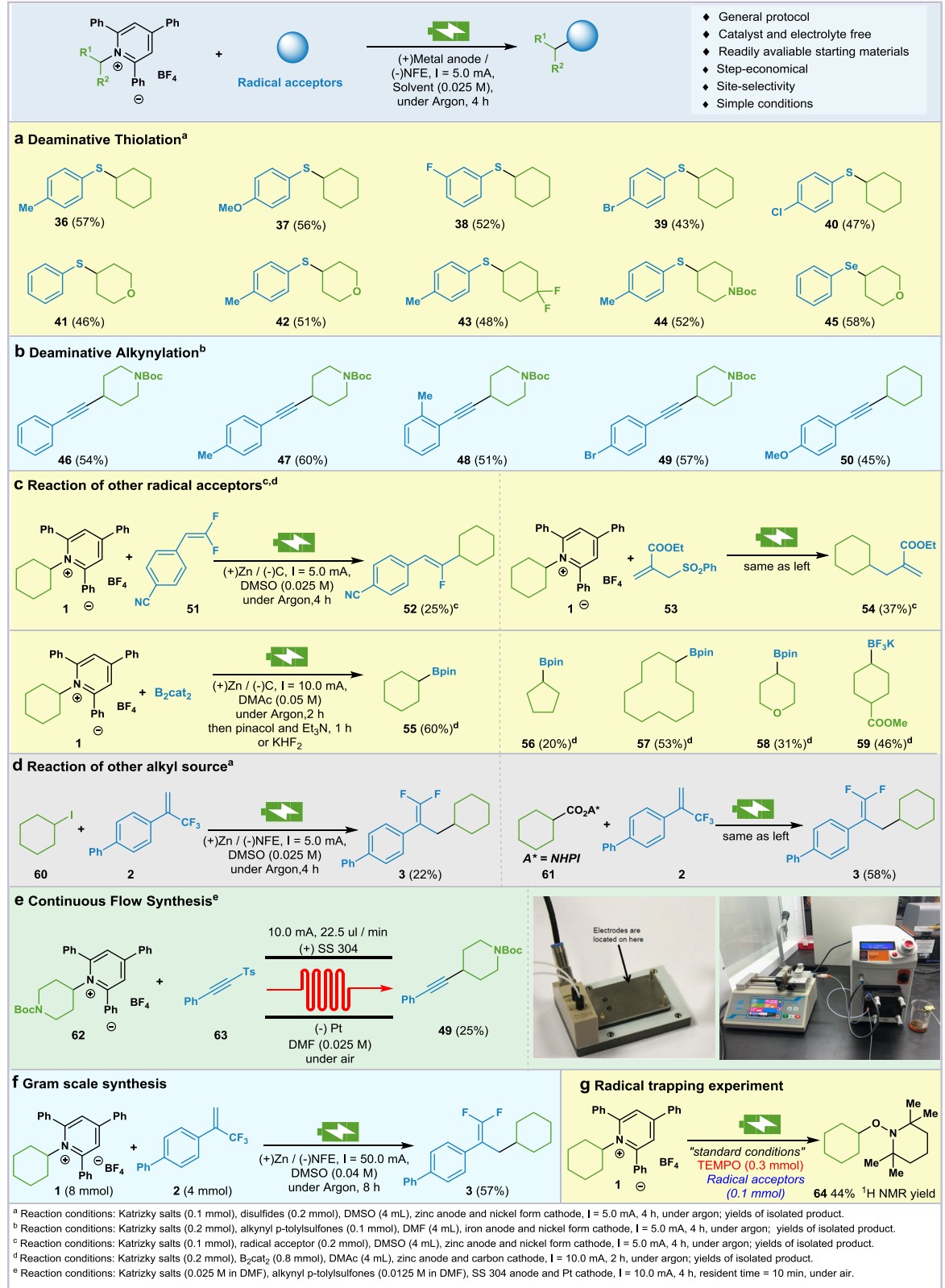

**Fig. 4 Scope of other radical acceptors and further elaboration. a** Deaminative thiolation. **b** Deaminative alkynylation. **c** Other radical acceptors. **d** Other alkyl source. **e** Microfluid synthesis. **f** Gram scale synthesis. **g** Radical trapping experiment.

**General procedure for the electrochemical reductive cross-coupling between alkylpyridinium salts and trifluoromethyl alkenes**. An oven-dried undivided reactor (10 mL) equipped with trifluoromethyl alkenes (0.1 mmol), alkylpyridinium salts (0.2 mmol) and a stir bar before adding DMSO (4 mL). The reactor was equipped with zinc electrode (52.5 × 8 × 2 mm) as the anode and foamed nickel electrode (52.5 × 8 × 2 mm) as the cathode. The reaction mixture was stirred and electrolyzed at a constant current of 5 mA (The dual display potentiostat was operating in constant current mode) under room temperature for 4 h. When the reaction was completed, the solution was extract by ethyl acetate (3 × 15 mL), and the combined organic layers were concentrated with a rotary

evaporator. The product was purified by flash column chromatography on silica gel.

**General procedure for the electrochemical reductive cross-coupling between alkylpyridinium salts and diaryl disulfides.** An oven-dried undivided reactor (10 mL) equipped with alkylpyridinium salts (0.1 mmol), diaryl disulfides (0.2 mmol) and a stir bar before adding DMSO (4 mL). The reactor was equipped with zinc electrode ($52.5 \times 8 \times 2$ mm) as the anode and foamed nickel electrode ($52.5 \times 8 \times 2$ mm) as the cathode. The reaction mixture was stirred and electrolyzed at a constant current of 5 mA (The dual display potentiostat was operating in constant current mode) under room temperature for 4 h. When the reaction was completed, the solution was extract by ethyl acetate ($3 \times 15$ mL), and the combined organic layers were concentrated with a rotary evaporator. The product was purified by flash column chromatography on silica gel.

**General procedure for the electrochemical reductive cross-coupling between alkylpyridinium salts and alkynyl p-tolylsulfones.** An oven-dried undivided reactor (10 mL) equipped with alkynyl p-tolylsulfones (0.1 mmol), alkylpyridinium salts (0.2 mmol) and a stir bar before adding DMF (4 mL). The reactor was equipped with iron electrode ($52.5 \times 8 \times 2$ mm) as the anode and foamed nickel electrode ($52.5 \times 8 \times 2$ mm) as the cathode. The reaction mixture was stirred and electrolyzed at a constant current of 5 mA (The dual display potentiostat was operating in constant current mode) under room temperature for 4 h. When the reaction was completed, the solution was extract by ethyl acetate ($3 \times 15$ mL), and the combined organic layers were concentrated with a rotary evaporator. The product was purified by flash column chromatography on silica gel.

**General procedure for the gram scale experiments.** An oven-dried undivided reactor (120 mL) equipped with trifluoromethyl alkenes 4 mmol, alkylpyridinium salts (8 mmol) and a stir bar before adding DMSO (100 mL). The bottle was equipped with zinc electrode ($30 \times 30 \times 1$ mm) as the anode and foamed nickel electrode ($30 \times 30 \times 1$ mm) as the cathode. The reaction mixture was stirred and electrolyzed at a constant current of 50 mA (The dual display potentiostat was operating in constant current mode) under room temperature for 8 h. When the reaction was completed, the solution was extracted with ethyl acetate ($3 \times 100$ mL), and the combined organic layers were concentrated with a rotary evaporator. The product was purified by flash column chromatography on silica gel.

## Data availability

All data supporting the findings of this study are available within the article and Supplementary Information files, or from the corresponding author upon request.

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

## Acknowledgements

We thank Collaborative Innovation Center of Advanced Microstructures and Jiangsu Provincial Key Laboratory of Photonic and Electronic Materials at Nanjing University for support. We gratefully acknowledge the financial support from the National Natural Science Foundation of China (Nos. 21772085 for Y.P., 21971107 for Y.P., and 22071101 for Y.W.).

## Author contributions

Methodology, Y.L.; Investigation, Y.L., X.T., Y.M., X.Y., L.K., and S.N.; Writing—review & editing, S.N., J.Q., and Y.W.; Supervision, Y.W., K.G., and Y.P.

## Competing interests

The authors declare no competing interests.
