## [Peer Review File · Nature Communications]

REVIEWER COMMENTS

Reviewer #1 (Remarks to the Author):

In this manuscript, the author developed a new method for deaminative reductive coupling of Katritzky salts with different radical acceptors, such as α -trifluoromethyl alkene, disulfide, diselenide, and alkynyl p-tolylsulfones under electrochemical conditions. This method features converting Katritzky salts to radical intermediates via single-electron reduction by cathode without using transition-metal catalysts or photocatalysts. Meanwhile, Katritzky salts could also serve as electrolytes under cell conditions. Furthermore, the author also applied continuous-flow technology to accelerate the reaction efficiency.

Though electrochemical platforms can access cost-effective, scalable, and distinctive transformations powered by inexpensive electricity, the reviewer believes using Katritzky salts to produce radical intermediate via electrochemistry is a straightforward simple idea without much rational design considering many available pieces of literature about Katritzky salts.

The reviewer suggests more control experiments to elucidate the reaction mechanism.

1. The author should perform more experiments to demonstrate the radical mechanism, such as the radical clock experiments. Currently, the reviewer can propose another pathway for the generation of compound 40 in table 2: the tempo was reduced first followed by a nucleophilic substitution.
2. The stoichiometric ratio of Katritzky salts and coupling partner is 2:1 in the absence of electrolyte, this phenomenon may be explained as one stoichiometric Katritzky salts serve as sacrificial electrolyte. Considering that frequently used electrolyte is much cheaper than reaction substrate, the author should give a compared result between no additional electrolyte reaction conditions and using additional electrolyte(s) in the presence of one stoichiometric Katritzky salts.
3. Since zinc was used as a sacrificial anode, the authors should demonstrate that Katritzky salts could not be reduced by zinc or zinc powder directly.

Additionally, the reviewer believes that the following issues should be solved:

1. The substrate scope of Katritzky salt is very narrow. Currently, it looks like that only a few simple amines are amenable to the reaction conditions.
2. The radical acceptors are all well known. The authors are recommended to disclose more interesting transformations using Katritzky salt in these electro-conditions.

Reviewer #2 (Remarks to the Author):

Electrochemical C–N Bond Activation for Deaminative Reductive Coupling of Katritzky Salts

Manuscript ID : 202102404

This manuscript reports electrochemical deaminative reductive coupling using Katritzky salts as radical precursors. In this work, electrochemistry was used instead of previously reported photocatalytic or transition metal catalytic conditions. This method offers an advantage for providing good reactivity through electrosynthesis as an efficient technique to promote single-electron reduction. In addition, this reaction has the advantage that no additional electrolyte is required because Katritzky salt can act as an electrolyte. On the other hand, this reviewer thinks this manuscript does not reach enough novelty and mechanistic interest for Nature Communication because precisely the same catalytic systems were used by previous works. For example, the reactions to generate gem-difluoro alkene from trifluoromethyl benzyl alkene has already been well developed by various groups such as Molander and Lei Zhou, and there is only a change in the coupling partner in the previously reported mechanism (For selected examples, *Angew. Chem. Int. Ed.* 2017, 56, 15073; *Asian J. Org. Chem.* 2019, 8, 661; *Chem. Sci.* 2020, 11, 10414.). In the current works, only the catalytic system was replaced as an extension of the previous research. Therefore, this reviewer thinks publication in OL

would be more appropriate.

Besides, please consider the following points.

1. The scope of Katritzky salt is very limited in the current works. Alkyl radicals showing reactivity are limited to secondary carbon radicals, and their functional group tolerability is also very limited. Please refer to the previous works (Angew. Chem. Int. Ed. 2017, 56, 12336, Chem. Eur. J. 2019, 25, 8240, ACS Catal. 2018, 11362, Org. Lett. 2019, 21, 2947, J. Am. Chem. Soc. 2018, 140, 10700.)
2. There is a large decrease in yield under air condition, and one may be curious about the reason.

Reviewer #3 (Remarks to the Author):

This manuscript describes the development of electrochemical conditions for the deaminative reactions of alkylpyridinium salts, which are efficiently made from alkyl amines. This report is important because it is the first example of using electrochemical conditions to activate alkylpyridinium salt that shows generality of the method. A clever aspect of this method includes the realization that addition of an electrolyte is unnecessary because the pyridinium salt serves this function. The scope is reasonably demonstrated, and three different partners are used, further demonstrating the generality of this approach.

I previously reviewed a version of this manuscript for a different journal. This revised manuscript is improved from the original version, and the effort the authors have taken is appreciated. I support its publication in Nature Communications after the following minor revisions have been accomplished:

- Martin's seminal example of using electrochemical reduction with alkylpyridinium salts (Org. Lett. 2019, 21 (8), 2947-2951. 10.1021/acs.orglett.9b01016) should be explicitly recognized. This could be simply adding a note to the beginning of that reference, such as, "A single example of a combined electrochemical/Ni catalyzed cross-coupling of alkylpyridinium salts has been demonstrated. See: ..."
- The authors need to address the fate of the excess pyridinium salt. Do they observe the formation of 200% triphenyl pyridine? Do they observe the addition of the alkyl radical to another equivalent of pyridinium salt?
- In the General Procedures, the authors need to clearly state when/how air was removed from the reaction vessel. Were these reactions set up in a glovebox? If so, when were they removed from the glovebox (before or after electrolysis)? If they were set up on the bench, when and how was air replaced with N₂? These details are absolutely necessary for others to reproduce this work and use this method.
- For the flow experiments, does yield continue to increase at slower flow rates?

Many thanks for reviewing our manuscript. We have carefully revised the article with point by point response to address the reviewers' comments.

Point by point response to the reviewers' comments:

Reviewer 1: In this manuscript, the author developed a new method for deaminative reductive coupling of Katritzky salts with different radical acceptors, such as α -trifluoromethyl alkene, disulfide, diselenide, and alkynyl p-tolylsulfones under electrochemical conditions. This method features converting Katritzky salts to radical intermediates via single-electron reduction by cathode without using transition-metal catalysts or photocatalysts. Meanwhile, Katritzky salts could also serve as electrolytes under cell conditions. Furthermore, the author also applied continuous-flow technology to accelerate the reaction efficiency.

Though electrochemical platforms can access cost-effective, scalable, and distinctive transformations powered by inexpensive electricity, the reviewer believes using Katritzky salts to produce radical intermediate via electrochemistry is a straightforward simple idea without much rational design considering many available pieces of literature about Katritzky salts.

The reviewer suggests more control experiments to elucidate the reaction mechanism.

1. The author should perform more experiments to demonstrate the radical mechanism, such as the radical clock experiments. Currently, the reviewer can propose another pathway for the generation of compound 40 in table 2: the tempo was reduced first followed by a nucleophilic substitution.

Response: Thank you for your comments. We have difficulties with the preparation of the cyclopropyl substrates for the radical clock experiments. Therefore, a radical capture experiment has been performed instead. Using diphenylethylene, the cyclohexal radical has been captured and detected by GC-MS (MW.262). To further demonstrate the possible reaction mechanism, electron paramagnetic resonance (EPR) experiments with N-tert-butyl- α -phenylnitron (PBN) as the electron-spin trapping reagent were performed. A significant EPR signal was observed for the model reaction, indicating a radical pathway.

i) PBN (No Signal)

ii) 1+2+PBN, (+)Zn / (-)NFE, I = 5.0 mA

2. The stoichiometric ratio of Katritzky salts and coupling partner is 2:1 in the absence of electrolyte, this phenomenon may be explained as one

stoichiometric Katritzky salts serve as sacrificial electrolyte. Considering that frequently used electrolyte is much cheaper than reaction substrate, the author should give a compared result between no additional electrolyte reaction conditions and using additional electrolyte(s) in the presence of one stoichiometric Katritzky salts.

Response: We have used less equivalents of Katritzky salts and common electrolyte such as $n\text{Bu}_4\text{NPF}_6$ for the reaction. Reduced yields have been obtained for the difluoroethylene products:

Entry	Katritzky salts	$n\text{Bu}_4\text{NPF}_6$	Yield of 3
1	2 equiv.	0	79
2	1 equiv.	0	51
3	1 equiv.	1 equiv.	50

3. Since zinc was used as a sacrificial anode, the authors should demonstrate that Katritzky salts could not be reduced by zinc or zinc powder directly.

Response: We have performed the reactions with zinc powder at 25 °C and lower yields have been obtained for difluoroethylene (**3**, 24%) and thiol ether (**21**, 20%).

Additionally, the reviewer believes that the following issues should be solved:

4. The substrate scope of Katritzky salt is very narrow. Currently, it looks like that only a few simple amines are amenable to the reaction conditions.

Response: We have used a variety of simple and complicated amines such as cyclohexylamine, piperidinamine, tetrahydrofuranamine and isobutanamine. Amino acids such as phenylalanine were also tolerated to give the deaminative coupling product **9** in 51% yield.

5. The radical acceptors are all well known. The authors are recommended to disclose more interesting transformations using Katritzky salt in these electro-conditions.

Response: Other radical acceptors such as difluoroethylenes and allyl sulfonates have also been applied to the standard conditions and the corresponding results were included in the revised manuscript (Table 2C).

Reviewer 2:

1. The scope of Katritzky salt is very limited in the current works. Alkyl radicals showing reactivity are limited to secondary carbon radicals, and their functional group tolerability is also very limited. Please refer to the previous works (Angew. Chem. Int. Ed. 2017, 56, 12336, Chem. Eur. J. 2019, 25, 8240, ACS Catal. 2018, 11362, Org. Lett. 2019, 21, 2947, J. Am. Chem. Soc. 2018, 140, 10700.)

Response: Thank you for your comments. We noticed a considerable amount of deaminative functionalizations have been reported. However, electrocatalysis has demonstrated great advantage for single electron transformation. The simple undivided cell and catalyst/electrolyte free platform offers excellent substrate tolerance in fluoroalkenylation, alkynylation and thiolation reactions. Furthermore, this globally compatible deaminative functionalization protocol can be facilitated by the rapid molecular diffusion across microfluidic channels. Therefore, we believe that this work is of great interest to the broad readership in the field of radical chemistry, electrochemistry and flow chemistry.

2. There is a large decrease in yield under air condition, and one may be curious about the reason.

Response: The decreased yields obtained in atmosphere was due to the sensitivity of cell system and difficulties in the isolation steps.

Reviewer 3:

1. Martin's seminal example of using electrochemical reduction with alkylpyridinium salts (Org. Lett. 2019, 21 (8), 2947-2951. 10.1021/acs.orglett.9b01016) should be explicitly recognized. This could be simply adding a note to the beginning of that reference, such as, "A single example of a combined electrochemical/Ni catalyzed cross-coupling of alkylpyridinium salts has been demonstrated. See: ..."

Response: Thank you for your comments. The related reference has been cited.

2. The authors need to address the fate of the excess pyridinium salt. Do they observe the formation of 200% triphenyl pyridine? Do they observe the addition of the alkyl radical to another equivalent of pyridinium salt?

Response: We have isolated triphenyl pyridine (170%) in the reaction mixture. No addition product of alkyl radical to another equivalent of pyridinium salt was found.

3. In the General Procedures, the authors need to clearly state when/how air was removed from the reaction vessel. Were these reactions set up in a glovebox? If so, when were they removed from the glovebox (before or after electrolysis)? If they were set up on the bench, when and how was air replaced with N₂? These details are absolutely necessary for others to reproduce this work and use this method.

Response: Thank you for your comments. The reaction mixture was prepared in the glove box and sealed under nitrogen atmosphere before moving out to the fumehood and a balloon filled with nitrogen was connected to the vessel.

4. For the flow experiments, does yield continue to increase at slower flow rates?

Response: At the rate of 11.25 $\mu\text{L}/\text{min}$, the flow experiment was not successful due to block of the microchannel. Therefore, a better flow equipment is required for further optimization.

Yours sincerely,

Yi Wang

Reviewers' comments:

Reviewer #1 (Remarks to the Author):

1. The substrate scope concerning Katrizky salt is still very narrow.
2. Using inexpensive electrolytes instead of Katrizky salt could also provide the desired product. This referee believes that the yield can be improved through optimization of the reaction conditions.

Reviewer #3 (Remarks to the Author):

I previously supported publication of this manuscript with minor changes. I continue to support its publication after following minor revisions have been accomplished:

- Another Reviewer asked about the efficiency of using Zn powder because Zn is used as the sacrificial anode. This is an important experiment that should be included in the Supporting Information and discussed in the manuscript. The potential that the sacrificial anode alone can do the reaction is an important question to answer, and these experiments contribute to an understanding of that.
- In their cover letter, the authors addressed my question about the fate of the excess pyridinium salt, stating that they observe 170% triphenylpyridine. This information should be added to the manuscript. Also, I did not see the result with 1 equiv of pyridinium salt. This experiment should be added to the Supporting Information.
- Related to the above point, the authors should also include their experiments using nBu₄NPF₆ as an electrolyte. This information was included in their cover letter, but not in the Supporting Information or manuscript.
- In their cover letter, the authors addressed my question about whether the reactions are set up in the glovebox. This should also be clearly stated in the General Procedures in the manuscript. These details are absolutely necessary for others to reproduce this work and use this method.
- Lines 78-79: "It provided a new route for the introduction of fluorine into amino acids" should be deleted. The deamination does not result in an amino acid product.
- I would also like to address the comments from the other reviewers about the limited scope in alkylpyridinium salt. I agree that this scope is somewhat limited, but also note that the requirement for 2 equiv of alkylpyridinium salt makes it unattractive to use expensive, complex alkyl amine precursors in this method. This limitation does detract from the direct impact of this method on synthesis, but I believe that this paper is still very high impact as the first example of electrochemical activation of pyridinium salts. It will certainly be highly cited as such.

Reviewer #1 (Remarks to the Author):

1. The substrate scope concerning Katrizky salt is still very narrow.

Response: Thank you for the comment. We have thoroughly investigated all available primary amines and most of the reported Katrizky salts up-to-date (Chem.Eur.J.2018, 24,17210; Org. Lett. 2019, 21, 2947; J. Am. Chem. Soc. 2019, 141, 16197; Org. Lett. 2019, 21, 3346; Angew. Chem. Int.Ed. 2019, 58,5697; Angew. Chem. Int. Ed. 2017, 56,12336; Adv. Synth. Catal., 2019, 361, 4902; ACS Catal. 2018, 8, 11362).

Both positive and negative results are shown below:

2. Using inexpensive electrolytes instead of Katrizky salt could also provide the desired product. This referee believes that the yield can be improved through optimization of the reaction conditions.

Response: The optimization for the electrolytes have been performed (SI, Table S5) and the results are shown below. 1.0 equiv. of Katrizky salt and electrolytes have been used and low to moderate yields have been obtained. Because a considerable amount of Katrizky salt resulted in the protonated by-product in the reaction mixture, excess Katrizky salt are necessary. Therefore, 1.0 equiv. of Katrizky salt is not enough, no matter which electrolyte was used.

Entry	Solvent	Electrolytes	Electrolytes	Electricity	Yield ^b
1	DMSO	Zn(+)/FNE(-)	ⁿ Bu ₄ NPF ₆	5 mA / 4 h	50
2	DMSO	Zn(+)/FNE(-)	ⁿ Bu ₄ NBF ₆	5 mA / 4 h	52
3	DMSO	Zn(+)/FNE(-)	LiClO ₄	5 mA / 4 h	46
4	DMSO	Zn(+)/FNE(-)	ⁿ Bu ₄ NBr	5 mA / 4 h	11
5	DMSO	Zn(+)/FNE(-)	ⁿ Bu ₄ NOAc	5 mA / 4 h	29
6	MeCN	Zn(+)/FNE(-)	ⁿ Bu ₄ NPF ₆	5 mA / 4 h	21
7	DMSO	Fe(+)/FNE(-)	ⁿ Bu ₄ NPF ₆	5 mA / 4 h	17
8	DMSO	Zn (+)/FNE(-)	ⁿ Bu ₄ NPF ₆	2.5 mA / 8 h	46
9	DMSO	Zn (+)/FNE(-)	ⁿ Bu ₄ NPF ₆	10 mA / 2 h	44
10 ^c	DMSO	Zn (+)/FNE(-)	ⁿ Bu ₄ NPF ₆	5 mA / 4 h	53

^aReaction conditions: 1a (0.1 mmol), 2a (0.1 mmol), solvent (4 mL), under argon atmosphere for 4 h. ^bDetermined by GC using 1,3,5-trimethoxybenzene as internal standard. ^c 50 °C

Reviewer #3 (Remarks to the Author):

I previously supported publication of this manuscript with minor changes. I continue to support its publication after following minor revisions have been accomplished:

- Another Reviewer asked about the efficiency of using Zn powder because Zn is used

as the sacrificial anode. This is an important experiment that should be included in the Supporting Information and discussed in the manuscript. The potential that the sacrificial anode alone can do the reaction is an important question to answer, and these experiments contribute to an understanding of that.

Response: Thank you for the comment. The Zn powder experiment has been discussed in the manuscript and the Supporting Information (Table 1, entry 13 and Table S2, entry 19). Without electricity, the sacrificial anode alone could result in trace amount of the product (Table 1, entry 12).

- In their cover letter, the authors addressed my question about the fate of the excess pyridinium salt, stating that they observe 170% triphenylpyridine. This information should be added to the manuscript. Also, I did not see the result with 1 equiv of pyridinium salt. This experiment should be added to the Supporting Information.

Response: The fate of the excess pyridinium salt has been included in the manuscript. The result with 1 equiv. of pyridinium salt has been included in SI Table S4.

- Related to the above point, the authors should also include their experiments using nBu₄NPF₆ as an electrolyte. This information was included in their cover letter, but not in the Supporting Information or manuscript.

Response: The nBu₄NPF₆ as electrolyte has been included in the manuscript (Table 1, entry 2).

- In their cover letter, the authors addressed my question about whether the reactions are set up in the glovebox. This should also be clearly stated in the General Procedures in the manuscript. These details are absolutely necessary for others to reproduce this work and use this method.

Response: The detailed procedure for the preparation of the reaction has been included in the SI.

- Lines 78-79: "It provided a new route for the introduction of fluorine into amino acids" should be deleted. The deamination does not result in an amino acid product.

Response: The related sentence has been deleted.

- I would also like to address the comments from the other reviewers about the limited scope in alkylpyridinium salt. I agree that this scope is somewhat limited, but also note that the requirement for 2 equiv of alkylpyridinium salt makes it unattractive to use expensive, complex alkyl amine precursors in this method. This limitation does detract from the direct impact of this method on synthesis, but I believe that this paper is still very high impact as the first example of electrochemical activation of pyridinium salts. It will certainly be highly cited as such.

Response: Thank you for the comment. Indeed, this electrochemical protocol does have limitation, it provides a new practical way for deaminative functionalizations. We have extended the scope in alkylpyridinium salt to improve the compatibility of this methodology.

REVIEWERS' COMMENTS

Reviewer #1 (Remarks to the Author):

Although the novelty of this method is moderate, in view of (i) the authors added more complex substrates in Table 1, and (ii) also the most important point, the author demonstrated that the Katritzky salt could undergo borylation in the electroconditions, I would support its publication in Nat. Comm. after following revisions have been accomplished:

(i) The grain size of Zn powder should be noticed—using 10 mesh zinc powder may give the product in 20% yield, while using 300 mesh zinc dust may give 80% yield.

(ii) The borylation reaction is the most attractive reaction in the revised manuscript, but the efficiency and the substrate scope of this reaction should be improved before publication.

Point by point response to the reviewers' comments:

Reviewer #1 (Remarks to the Author):

Although the novelty of this method is moderate, in view of (i) the authors added more complex substrates in Table 1, and (ii) also the most important point, the author demonstrated that the Katritzky salt could undergo borylation in the electroconditions, I would support its publication in Nat. Comm. after following revisions have been accomplished:

(i) The grain size of Zn powder should be noticed—using 10 mesh zinc powder may give the product in 20% yield, while using 300 mesh zinc dust may give 80% yield.

Response: Thank you for your comments. The Zn powder we used were 40-50um (300-400 mesh). We also tested 200, 800 and 1600 mesh, the products were obtained in lower yields (<20%).

(ii) The borylation reaction is the most attractive reaction in the revised manuscript, but the efficiency and the substrate scope of this reaction should be improved before publication.

Response: Thank you for your comments. Considering the deaminative borylation has been well-studied (J. Am. Chem. Soc., 2018, 140, 10700; Angew. Chem. Int. Ed. 2018, 57, 15227), we only showed a few examples to demonstrate the possibility of such process under electrochemical conditions.

Yours sincerely,

Yi Wang